# Micro-size plankton abundance and assemblages in the western North Pacific Subtropical Gyre under microscopic observation

**Taketoshi Kodama[ID]\*, Tsuyoshi Watanabe, Yukiko Taniuchi, Akira Kuwata, Daisuke Hasegawa**

Fisheries Resources Institute, Japan Fisheries Research and Education Agency, Yokohama, Japan

\* takekodama@affrc.go.jp

**Data Availability Statement:** All data are available from Mendeley Data (http://dx.doi.org/10.17632/hmy5jzccyx.2).

## Abstract

While primary productivity in the oligotrophic North Pacific Subtropical Gyre (NPSG) is changing, the micro-size plankton community has not been evaluated in the last 4 decades, prompting a re-evaluation. We collected samples over three years (2016–2018) from depths of 10 to 200 m ($n = 127$), and the micro-size plankton were identified and counted to understand the heterogeneity of micro-size plankton community structure. The assemblages were consistent to the those of 4 decades ago. Dinophyceae (dinoflagellates) were the most numerically abundant, followed by Cryptophyceae and Bacillariophyceae (diatoms). The other micro-size plankton classes (Cyanophyceae, Haptophyceae, Dictyochophyceae, Euglenophyceae, and Prasinophyceae) were not always detected, whereas only *Trichodesmium* spp. was counted in the Cyanophyceae. Other unidentified autotrophic and heterotrophic flagellates were also significantly present, and their numeric abundance was higher than or at the same level as was that of the Dinophyceae. In the Dinophyceae, *Gymnodiniaceae* and Peridiniales were abundant. The chlorophyll *a* concentration and these class-level assemblages suggested micro-size plankton is not a major primary producer in this area. We applied generalized additive models (GAMs) and principal coordination analyses (PCoAs) to evaluate the habitats of every plankton group and the heterogeneity of the assemblages. The GAMs suggested that every classified plankton abundance showed a similar response to salinity, and we observed differences in habitats in terms of temperature and nitrate concentrations. Based on the PCoAs, we observed unique communities at the 200 m depth layer compared with those at the other sampling layers. The site scores of PCoAs indicated that the micro-size plankton assemblages are most heterogeneous at the 10 m depth layer. At such depth, diazotrophic Cyanophyceae (*Trichodesmium* spp.) are abundant, particularly in less-saline water. Therefore, nitrogen fixation may contribute to the heterogeneity in the abundance and assemblages in the western NPSG.

**Funding:** This work was supported by grants from the Project of the Bio-oriented Technology Research Advancement Institution, NARO (the special scheme project on advanced research and development for next-generation technology) to all. The funder did not play any role in neither the study design, data collection and analysis, decision to publish, nor preparation of the manuscript.

**Competing interests:** No authors have competing interests.

# Introduction

The size of primary producers is one of the keys to controlling the productivity of marine ecosystems [1]. In waters where pico-size phytoplankton are dominant, more trophic levels are required to convert primary production to useful forms; therefore, biomass transfer efficiency is low [1]. In other words, energy fixed by primary production is more efficiently transferred to higher trophic organisms in micro-plankton dominated waters [1]. In addition, over 100% of daily primary production is sometimes consumed by micro-size heterotrophic and mixotrophic plankton [2, 3]. Consequently, the abundance and composition of micro-size plankton comprise necessary information for evaluating the biological productivity of the ocean.

The subtropical open ocean occupies approximately 40% of the surface of the Earth. The stratification of the surface water column develops throughout the year, and nutrient supplies from the deeper layer are extremely limited; therefore, phytoplankton abundance is low and limited in this layer. It has been considered that the ecosystem of the subtropical open ocean is stable and is a 'climax-type' community [4]. However, over the recent three decades, many studies have indicated the ecosystem of the subtropical open ocean as unstable and heterogeneous in space and time [5]. In the subtropical open ocean, pico- and nano-size phytoplankton is the dominant primary producers; however, micro-size phytoplankton is also present in significant numbers [5–7]. In particular, diatoms (Bacillariophyceae) are sometimes abundant ($> 10^8$ cells m$^{-2}$) in the surface mixed layer [5, 8, 9], with most being nitrogen-fixing diatom symbioses [10].

The North Pacific Subtropical Gyre (NPSG), the widest gyre in the ocean, is sometimes divided into two provinces, namely west and east [11]. In the western NPSG, both nitrate and phosphate are depleted at nanomolar levels [12], and nitrogen fixation is significant, but not quite active [13]. The western NPSG is oligotrophic, but is known as the spawning and nursery grounds of commercially important migratory fish, such as Albacore (*Thunnus alalunga*), Skipjack tuna (*Katsuwonus pelamis*), Blue marlin (*Makaira mazara*), and Japanese eel (*Anguilla japonica*) [14, 15]. In the western NPSG, the micro-size plankton community was studied widely in the 1970s [6, 16]. The findings indicated that heterotrophic protists such as flagellates and dinoflagellates are dominant, with autotrophic coccolithophore and diatoms also significantly present. These communities are similar to those of the eastern NPSG [7]. However, with the recent effects of global warming, primary productivity has been changing, even in the nutrient-depleted oligotrophic ocean [17, 18]. Therefore, the micro-size plankton community in the western NPSG could have changed recently compared with the findings of the studies in the 1970s. In addition, descriptions of the classical morphologically identified plankton community are necessary for further studies using state-of-the-art technology, such as the metabarcoding technique.

In view of the above, we investigated the morphologically identified micro-size plankton community structure in the western NPSG. We evaluated the habitats of plankton groups by using empirical statistical models (generalized additive models, GAMs) and multivariate analyses (principal coordination analysis, PCoA). Our aim was to add to the basic knowledge on the micro-size plankton community structure in the oligotrophic western NPSG to understand the trophic structure and heterogeneity of this community.

# Materials and methods

## Sampling

We collected samples during three cruises on the R/V *Kaiyo-maru* (Japan Fisheries Agency) conducted in September and October 2016 (named the KY1604 cruise), September and

**Table 1. List of the sampling layers and numbers in three cruises.**

| Cruise | Date | Sampling layers | Numbers of samples |
|---|---|---|---|
| KY1604 | 28th Sep. –31st Oct., 2016 | 10 m | 9 |
| | | 50 m | 9 |
| | | 100 m | 9 |
| | | 200 m | 9 |
| | | SCM (100–130 m) | 9 |
| KY1704 | 29th Sep.–25th Oct., 2017 | 10 m | 5 |
| | | 50 m | 5 |
| | | 100 m | 32 |
| | | 200 m | 5 |
| | | SCM (85–105 m) | 5 |
| KY1801 | 2nd Jun. –5th Jun., 2018 | 10 m | 6 |
| | | 200 m | 6 |
| | | SCM-High (85–120 m) | 6 |
| | | SCM (125–155 m) | 6 |
| | | SCM-Low (150–175 m) | 6 |

The SCM, SCM-High and SCM-Low denote subsurface chlorophyll maximum (the peak of in vivo chlorophyll fluorescence), the shallower and deeper SCM edges (approximately half of fluorescence values compared with the SCM), respectively.

October 2017 (KY1704 cruise), and June 2018 (KY1801 cruise). The cruises were conducted in the western NPSG between 12˚N and 25˚20′N and between 126˚E and 143˚E (Table 1, Fig 1). The primary aim was investigating the distribution of Japanese eel larvae; therefore, the observations for collections of seawater were limited to night-time collections.

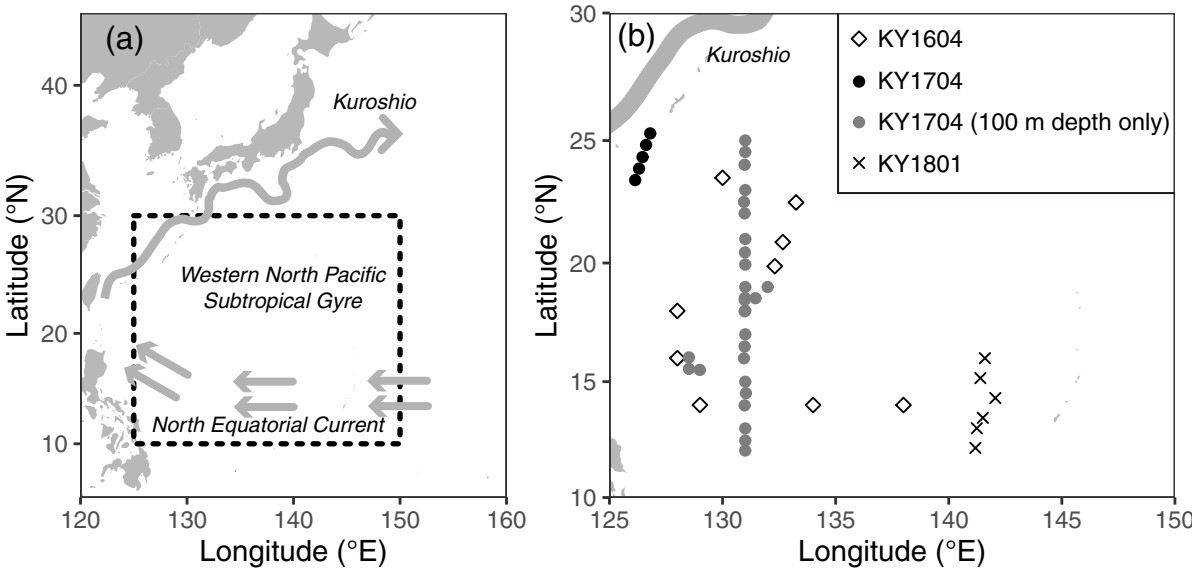

**Fig 1. Sampling locations of this study.** (a) Small-scale map of the western North Pacific, and (b) enlarged map of the sampling sites during the three cruises. The gray arrows in (a) denote schematic major ocean currents in the observation areas, but the Kuroshio current path is that of October 2017 based on the *Quick Bulletin of Ocean Conditions* issued by the Japan Coast Guard. (https://www1.kaiho.mlit.go.jp/KANKYO/KAIYO/qboc/index_E.html).

All observation and samplings during the cruises were permitted by Japan Fisheries Agency (SUISUI NO. 27–1253, NO. 29–28, and NO. 30–238). Our samplings were conducted in high seas or Japanese EEZ area during KY1604 and KY1704 cruises, and those during KY1801 cruise were permitted by the coastal States (the Federated States of Micronesia and United States of America) as the marine scientific research (MSR) in accordance with the United Nations Convention on the Law of the Sea (UNCLOS). The permission during KY1801 cruise was granted by Japan Fisheries Agency (No. U2018-001).

The samples for identifying micro-sized plankton abundance and assemblages were collected using a conductivity-temperature-depth profiler with carousel multiple sampling (CTD-CMS), which consists of a CTD sensor (SBE911plus, Sea-Bird Scientific, USA) and a Sea-Bird carousel water sampler (SBE32 with 10-litter Niskin-X sampling bottles). Chlorophyll fluorescence was monitored using a fluorometer (Seapoint Sensors, USA) attached to the CTD sensor. Water samples were collected at 10, 50, 100, and 200 m depths, and the subsurface chlorophyll maximum (SCM) layer at nine stations during the KY1604 cruise (Table 1). During the KY1704 cruise, the same vertical profiles were conducted at five stations in the northwestern part of the study area. At the other 28 stations, the samples were collected only at 100 m depth during the KY1704 cruise (Table 1). We did not collect a SCM sample when it was 100 m at one station during the KY1704 cruise. At six stations during the KY1801 cruise, the water samples were collected at depths of 10 and 200 m, and two edges and peak of the SCM to understand the variations in the SCM layers (Table 1). The shallower and deeper SCM edges (approximately half of fluorescence values compared with the SCM) were named SCM-High and SCM-Low, respectively. In total, 127 samples were collected during three cruises for analyses of the phytoplankton community structure.

For plankton assemblages and abundance analyses, 40 ml of acidified Lugol's solution was added immediately to 1 L seawater (4% final concentration). The sample treatments were similar to those of Watanabe et al. [19]. The samples were kept in the dark at room temperature (~20°C) until the onshore treatments. Samples were concentrated by reverse filtration through 2 μm membrane filters [20]. Concentrated algal cells were settled in a chamber and counted using an inverted microscope [21]. Because we used acidified Lugol's solution as the fixing reagent, some plankton groups with calcium carbonate plates, such as coccolithophores, were not detected in our samples. We counted plankton classified into eight classes, namely Bacillariophyceae (diatoms), Cyanophyceae, Cryptophyceae, Dinophyceae (dinoflagellates), Dictyochophyceae, Euglenophyceae, Haptophyceae, and Prasinophyceae. Bacillariophyceae, Cyanophyceae, Dinophyceae, Dictyochophyceae, Haptophyceae, and Prasinophyceae were classified mostly into genus level when possible. The identification of Cryptophyceae and Euglenophyceae remained at class level. The other autotrophic and heterotrophic flagellates, except eight classes, were not identified, but were counted (hereafter, described as flagellates). The phytoplankton species were identified following Tomas et al. [22]. *Trichodesimum* spp. is the only counted group in Cyanophyceae, with the other micro-size Cyanobacteria, such as *Richelia intracellularis*, not counted from these samples. In the case of *Trichodesimum* spp., the number of filaments were counted, but not that of cells. During the KY1704 and KY1801 cruises, diatom–diazotroph (*R. intracellularis*) associations (DDAs) were counted from the specimens collected on 10 μm filtered 2.3L of seawaters through a 10 μm filter following Chen et al. [23].

We also collected samples for measurements of nutrients and chlorophyll *a* concentration. Approximately 10 mL of water was collected for nutrients, and kept frozen at -20°C until onshore analysis. The samples were thawed >24 h before analysis for recovering from polysilicate to monosilicate, after which the nutrient (nitrate, nitrite, phosphate, and silicate) concentrations were investigated using an autoanalyzer (QuAAtro, BLTEC, Japan). The

measurement protocols depended mainly on information from Becker et al. [24]. For each chlorophyll (Chl) a sample, 300 mL of water was collected. The particles in the water were collected on four types of filter, namely a glass fiber filter (GF/F, Whatman, 0.7 μm mesh, Merck, Germany), 0.2 μm membrane filter (Nuclepore, Whatman, Merck, Germany), 3.0 μm membrane filter (Nuclepore, Whatman, Merck, Germany), and 10 μm membrane filter (Nuclepore, Whatman, Merck, Germany). The filters were immersed in N,N-dimethylformamide [25] and stored below -20˚C in the dark until onshore measurements. Chl a concentrations were measured with a Turner Designs 10-AU fluorometer (USA) [26]. In this study, we used the results of the Whatman GF/F filter (as total Chl a concentration) and the Whatman Nuclepore 10 μm filter (10 μm Chl a concentration, hereafter). During the KY1604 cruise, the 10 μm Chl a concentration was lacking, except at the SCM layer.

As several previous studies have indicated that phytoplankton abundance and assemblages differ with the eddy structure (cyclonic or anticyclonic) in the open ocean [27–29], we used relative vorticity (Ro) as the index of the eddy structure. Ro is calculated from the following equations

$$\text{Ro} = \zeta/f \tag{1}$$

where $\zeta$ is absolute vorticity, and $f$ is the Coriolis parameter. The parameters for each sampling point were calculated from absolute geostrophic velocities, and geostrophic velocity anomalies from the sea level daily gridded data product (Product identifier: SEALEVEL_GLO_-PHY_L4_REP_OBSERVATIONS_008_047) of the European Union Copernicus Marine Environment Monitoring Service. The positive and negative Ro indicated the position in the cyclonic and anticyclonic eddy, respectively.

## Statistical analyses

All the statistical analyses were performed with R software [30]. We applied two statistical analyses for evaluating the relationship between the micro-size plankton community and environmental parameters. One is the generalized additive model (GAM) analysis, which is one of the empirical models for understanding the ecological habitats of target biota [31], and the other is PCoA, which is one of the classical multidimensional scaling techniques [32, 33]. We chose PCoA because our data sets did not meet the prerequisites for principal component analysis (PCA).

GAM analysis was applied to the cell density of every plankton group classified at class level, except flagellates, which were not identified into class level. We applied GAM using the *gam* function in the mgcv package [34] to examine the influence of oceanographic variables (i.e., temperature, salinity, nitrate, and silicate concentration) and sampling variables (i.e., latitude). We used a gamma distribution and a natural log link function to model plankton density. The full model is:

$$P_i \sim s(\text{T}) + s(\text{S}) + s(\text{Lat}) + s(\text{logNO3}) + s(\text{Ro}) + \beta \tag{2}$$

where $P_i$ represents the cell density of target plankton groups ($i$). T, S, Lat, logNO3, Ro, and $\beta$ represent temperature, salinity, latitude, common log-transformed nitrate concentration, relative vorticity, and intercept, respectively. $s$ is a spline function. The upper limits of the degrees of freedom for all smoothing terms ($k$) were set to 4 to avoid biologically impossible responses [35]. Multicollinearity in the environmental factors was checked using the variance inflation factor (VIF), and all VIF were <5 [36]. As regards the results, we did not include depth, which transformed as the categorical values (10 m, 50 m, 100 m, SCM, and 200 m), and the silicate and phosphate concentration in the model. We chose the optimal GAMs based on a corrected Akaike information criterion (AICc) value.

When the target taxon was absent in >30% of samples, we applied the delta-GAM technique. First, the target cell density data were transformed as present (1) or absent (0), and we applied GAMs with a binomial distribution and a logit link function (hereafter, present–absent GAM). Subsequently, after removing the absent sample data, GAM was re-performed with the gamma distribution and the natural log link function to model the target plankton cell density (hereafter, abundance GAM). The explanatory variables in the full model descriptions of the delta-GAM were the same with Eq 2. When the target taxon was absent in >80% of samples, we did not apply any GAMs to them.

The PCoA was used to evaluate the spatial heterogeneity of plankton assemblages. We applied PCoA to three types of communities, namely 1) classified into class level including the flagellates, 2) genus level data of the Dinophyceae, and 3) genus level data of the Bacillariophyceae. Our data contained significant null data; therefore, the distance among the samples were calculated with Bray–Curtis dissimilarity [33]. PCoA was conducted using the *cmdscale* function in the VEGAN package [37]. PCoA outputs scores for samples. The scores for target communities were calculated by the weighted average values. The relationships between environmental parameters and PCoA output scores for stations were calculated with the *envfit* function in the VEGAN package [37]. The environmental parameters were limited to those used in the GAMs.

## Results

### Environmental conditions

The SCM layer was formed at 115 ± 9 m (mean ± SD) in the KY1604, 94 ± 10 m in the KY1704, and 138 ± 10 m in the KY1801 cruises. SCM-High and SCM-Low in the KY1801 cruise were set at 102 ± 13 and 166 ± 8 m, respectively. Temperature decreased with depth; however, the temperature at 10 m was indicated the same as that at 50 m depth in the KY1704 cruise (Fig 2). Less-saline water was observed at 10 m depth during every cruise and, in particular, it was <34.5 at 10 m depth in the KY1801 cruise (Fig 2). The nitrate concentration was largely depleted from 10 m to the SCM, and remained at several µM at 200 m depth (Fig 2). Nitrate concentration was over 1 µM of in water where the temperature was below 20–25˚C. The vertical distribution of silicate and phosphate concentrations was quite similar to the nitrate concentration, while the silicate concentration was not depleted <1 µM, even at a depth of 10 m. The total Chl *a* concentration was highest at the SCM or SCM-High layers, suggesting the fluorometer values were overestimating the chlorophyll *a* concentration in the deeper layers as indicated by Falkowski and Kiefer [38]. The total Chl *a* concentration was lowest at 200 m depth (Fig 2). The median values of 10 µm Chl *a* concentration were highest at SCM (Fig 2), but the contribution to the total Chl *a* concentration was limited to <5.0% in the median. The contribution to the total Chl *a* concentration increased at a depth of 200 m (Fig 2). In addition, the contribution of 10 µm Chl *a* was high at 10 m depth in the KY1801 cruise (Fig 2).

### Micro-size plankton abundance and assemblages

The layer of micro-sized plankton cell density varied widely from $1.1 \times 10^3$ to $7.3 \times 10^4$ cells L$^{-1}$ (mean ± SD: $1.4 ± 1.3 \times 10^4$ cells L$^{-1}$), and the vertical profiles differed among the cruises (Fig 3). In KY1604, the cell density was highest at the SCM ($9.8 ± 3.4 \times 10^3$ cells L$^{-1}$) and lowest at 200 m depth ($3.3 ± 1.9 \times 10^3$ cells L$^{-1}$). In the KY1704 cruise, the mean cell density was highest at 100 m depth ($2.4 ± 1.6 \times 10^4$ cells L$^{-1}$) and the next highest at 50 m depth ($1.3 ± 0.6 \times 10^4$ cells L$^{-1}$, Fig 3). In the KY1801 cruise, the total cell densities decreased with depth, with the mean cell abundance being highest at 10 m depth ($3.7 ± 1.0 \times 10^4$ cells L$^{-1}$).

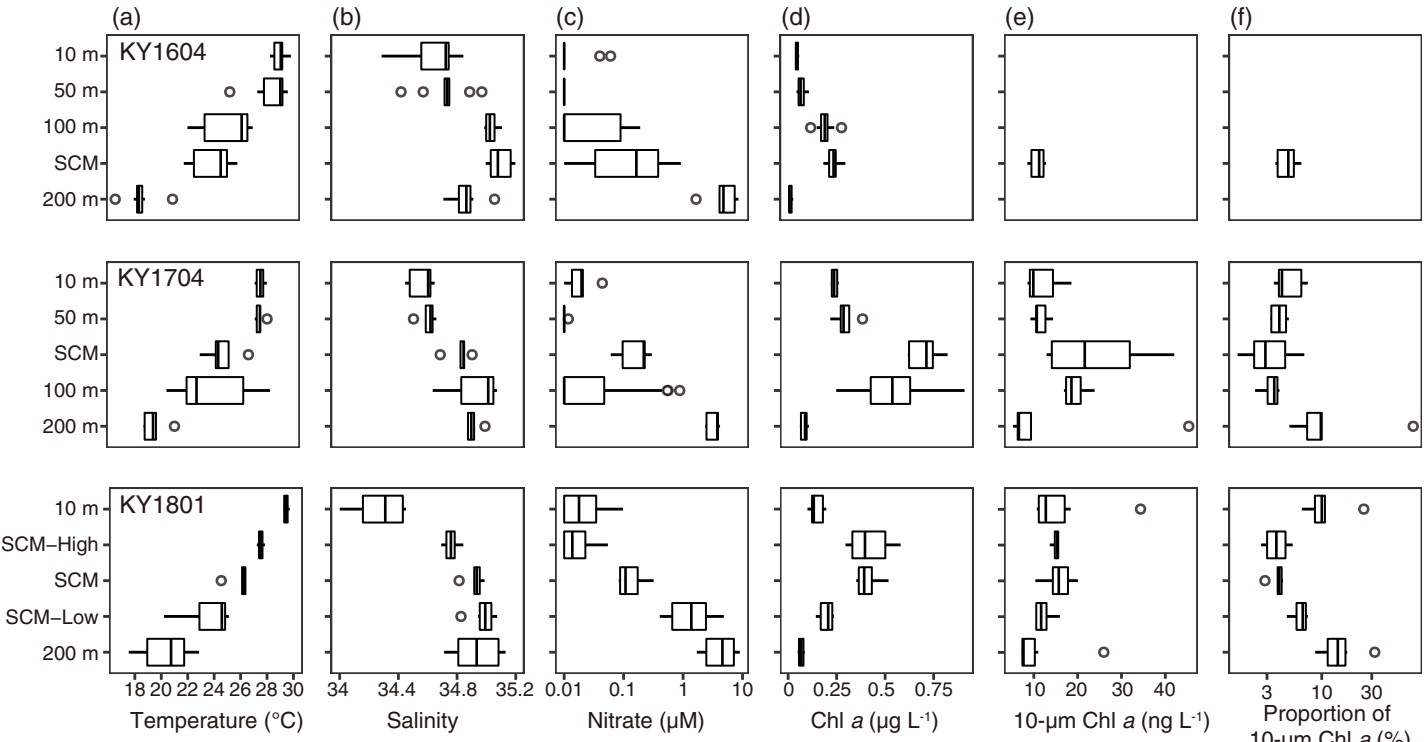

**Fig 2.** Vertical distribution of (a) temperature, (b) salinity, (c) nitrate concentration, (d) total Chl *a* concentration, (e) 10 μm Chl *a* concentration, and (f) proportion of 10 μm Chl *a* concentration to total Chl *a* concentration for KY1604 (top), KY1704 (middle), and KY1801 (bottom). The thick vertical line in each box represents the median values. Boxes indicate the lower and upper quartiles. Horizontal lines extending from each box represent the minimum and maximum values. The open circles are outliers. On the nitrate concentration and proportion of 10 μm Chl *a* concentration, the x-axes are transformed as common logarithms.

At the class level including flagellates, Dinophyceae (total mean ± SD: $6.2 ± 5.0 ×10^4$ cells $L^{-1}$) and flagellates ($6.2 ± 6.6 ×10^4$ cells $L^{-1}$) were the dominant groups in all the samples. They were dominant in at least 62.6% of total cells (mean ± SD: 88.6 ± 7.4%). The third and fourth most dominant groups were the Cryptophyceae ($1.0 ± 2.0 × 10^3$ cells $L^{-1}$) and Bacillariophyceae ($0.6 ± 0.5 × 10^3$ cells $L^{-1}$). These findings were similar among the cruises and depths, but the contributions of Cryptophyceae and Bacillariophyceae were slightly higher below depths of 100 m (Fig 3). The other five groups were rare (mean abundance was <1% of the total cell density). Prasinophyceae was the fifth most dominant class on average. *Pyramimonas* spp. was the only identified in the Prasinophyceae and detected from 69 of 127 samples. Dictyochophyceae, only *Dictyocha* spp. identified and detected, was observed in 81 of 127 samples, but the numeric contribution was <2.7%. *Trichodesmium* spp. was the only counted in the Cyanophyceae class, which contributed <0.5% to the numerical base and was detected in 33 of 127 samples. In the glutaraldehyde-fixed samples, but limited to 10 m depth and SCM layers, DDAs (*Richelia-Hemiaulus*, *Richelia-Rhizosolenia*, and *Richelia-Chaetoceros* symbioses) were always <70 host cells $L^{-1}$ at 10 m depth, and <9 host cells $L^{-1}$ at the SCM layer. Haptophyceae (only detected *Phaeocystis* spp.) were observed only in two samples, but were dominant at 19.8% and 28.8% of total plankton cells in the samples. Euglenophyceae were observed only in three samples, with negligible contribution (<1.1% of total cells). Detailed vertical distributions of plankton density at class level showed that the KY1801 cruise differed from the other two cruises (Fig 4). In the KY1604 and KY1704 cruises, the median values of densities of planktons, except Cyanophyceae (*Trichodesmium* spp.), were highest at SCM or 100 m depth (Fig 4). The abundance of *Trichodesmium* spp. was highest at 10 m and 50 m depth in the KY1604 and

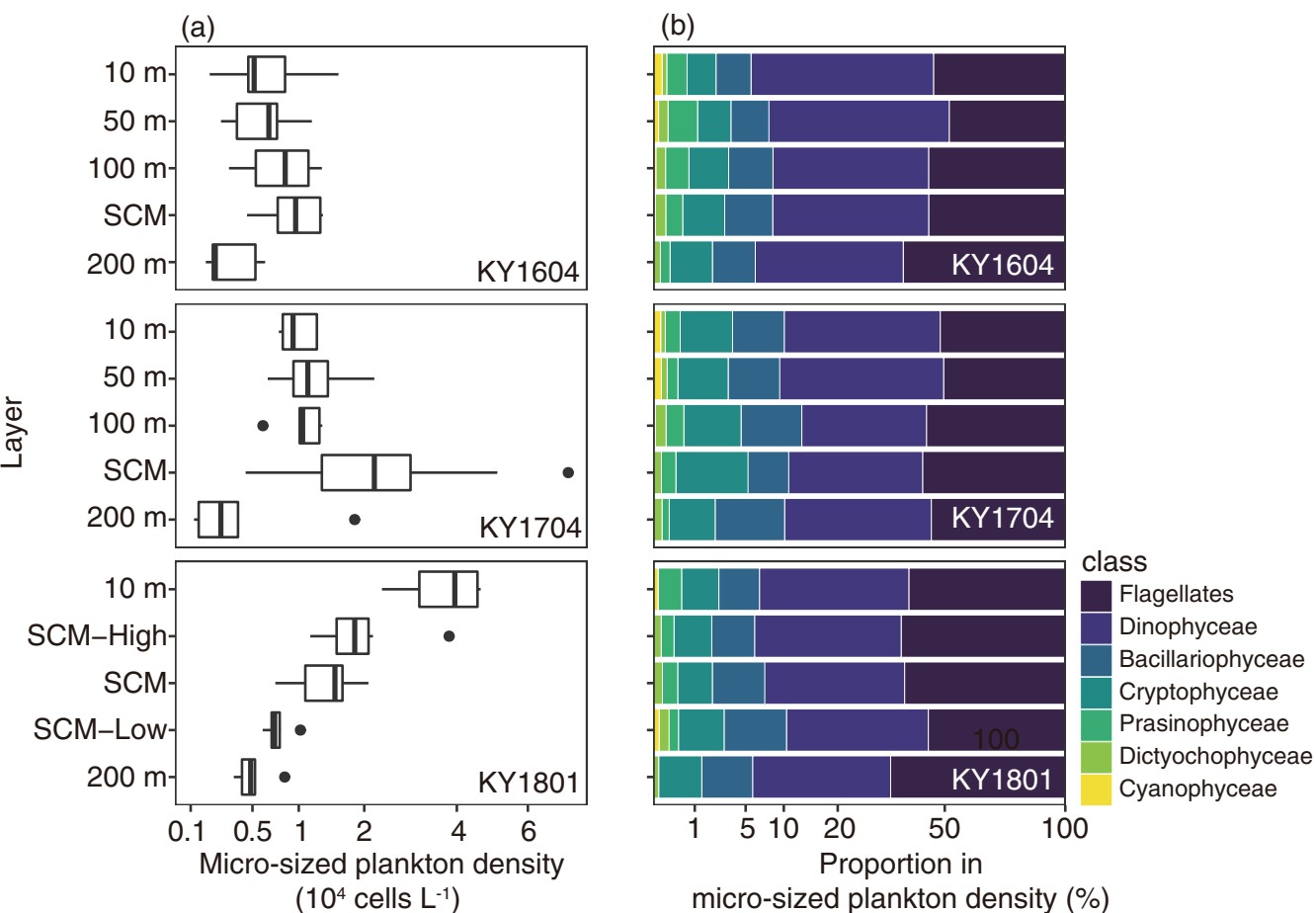

**Fig 3.** Vertical distribution of (a) micro-size plankton density (cells L$^{-1}$) and (b) their mean community structure for KY1604 (top), KY1704 (middle), and KY1801 (bottom). Haptophyceae and Euglenophyceae are not present in the community structure panel. In the boxplots, the thick vertical line in each box represents the median values. Boxes indicate the lower and upper quartiles. Horizontal lines extending from each box represent the minimum and maximum values. The closed circles are the outliers. The x-axes are transformed square root values.

KY1704 cruises, respectively. In contrast with the KY1604 and KY1704 cruises, the abundances, except Dictyochophyceae (*Dictyocha* spp.) and Cryptophyceae, were highest at 10 m depth in the KY1801 cruise.

Dinophyceae were detected in 21 genera, one family (Gymnodiniaceae), and one order (Peridiniales). The contributions of Gymnodiniaceae and Peridiniales were large (Fig 5), with the cell numbers of Gymnodiniaceae dominant at 26.2–94.4% of Dinophyceae (mean 70.8%), and Peridiniales was dominant at 0–58.4% (mean 14.9%). When the mean contribution was calculated with every cruise and layer, only six genera were recorded >1% mean contribution of Dinophyceae (*Gyrodinium*, *Oxytoxum*, *Pronoctiluca*, *Prorocentrum*, *Scrippsiella*, and *Heterocapsa*, Fig 5). In the KY1604 cruise, *Oxytoxum* was abundant at genus level from 10 m to the SCM layer, and in the other two cruises, *Heterocapsa* was dominant or at the same level as *Oxytoxum* from 10 m to the SCM layer (Fig 5). At 200 m depth, *Pronoctiluca* was abundant for all three cruises (Fig 5).

Bacillariophyceae were detected in 34 genera, all identified into genus level. The vertical distribution of the composition of Bacillariophyceae in the three cruises are shown in Fig 5. The genus of which the contributions were always <5% of the total Bacillariophyceae in every cruise and layer is treated as the other Bacillariophyceae. The contribution to the total

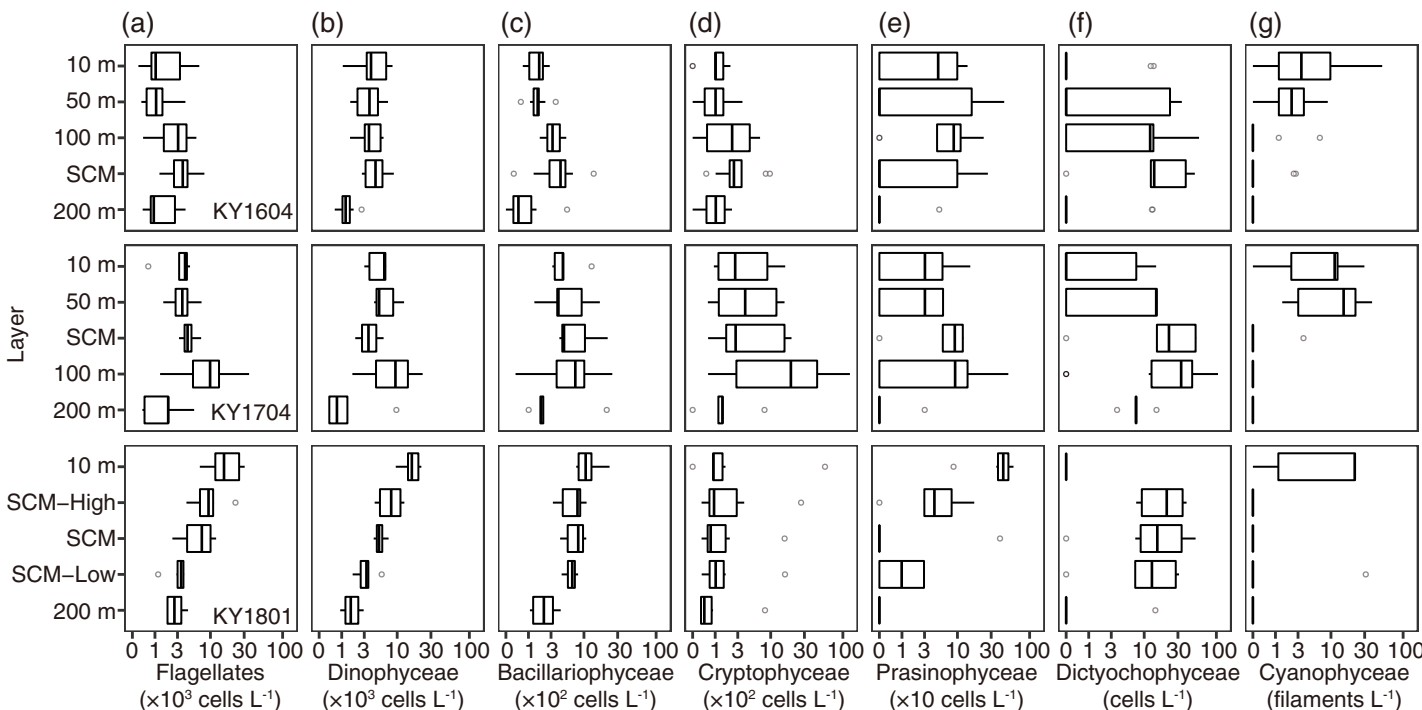

**Fig 4.** Vertical distributions of densities of the seven major plankton groups, i.e., (a) flagellates, (b) Dinophyceae, (c) Bacillariophyceae, (d) Cryptophyceae, (e) Prasinophyceae, (f) Dictyochophyceae, and (g) Cyanophyceae (*Trichodesmium* spp.) for KY1604 (top), KY1704 (middle), and KY1801 (bottom) cruises. The thick vertical line in each box represents the median values. Boxes indicate the lower and upper quartiles. Horizontal lines extending from each box represent the minimum and maximum values. The open circles are outliers. The x-axes are transformed logarithm values.

Bacillariophyceae abundance differed among the layers (Fig 5), at depths of 10 and 50 m, *Navicula* and *Nitzschia* were dominant on average and, below the SCM, *Fragilariopsis* was dominant on average. *Pseudo-nitzschia* was significantly present below 100 m depth. *Cylindrotheca* was also significantly present at 100 m and the SCM layer in the KY1604 and KY1704 cruises and significantly present at 10 m depth in the KY1801 cruise. *Thalassiosira* was sometimes abundant >20% of Bacillariophyceae (e.g., at 200 m in the KY1704 cruise).

## Response to environmental parameters based on GAMs

Because the Haptophyceae and Euglenophyceae were rarely detected in the samples, we did not perform GAMs on these two classes. Flagellates, Dinophyceae, Bacillariophyceae, and Cryptophyceae were detected and observed from >70% of samples, and we performed the GAMs only on their abundance. The model descriptions and deviance explained values are presented in Table 2. The full model was the least-AICc model in the case of Dinophyceae and Cryptophyceae and, in the other two groups, Ro was removed from the optimal model (Table 2). In other words, temperature, salinity, latitude, and nitrate concentration were selected as the explanatory variables in all the optimal models of these four groups (Table 2). While Ro remained in the optimal models in the cases of Dinophyceae and Cryptophyceae, the response was not significant ($p > 0.05$).

Temperature showed unimodal effects (Dinophyceae and Bacillariophyceae) or monotonical negative effects with warming (Dinophyceae and Cryptophyceae) to the abundance of every group (Fig 6). Here, the spline function indicated the relationship between the target group abundance and the target parameter after the other parameters were fixed, and the additive effect 0 was set as the mean abundance of targets. In other words, the negative additive

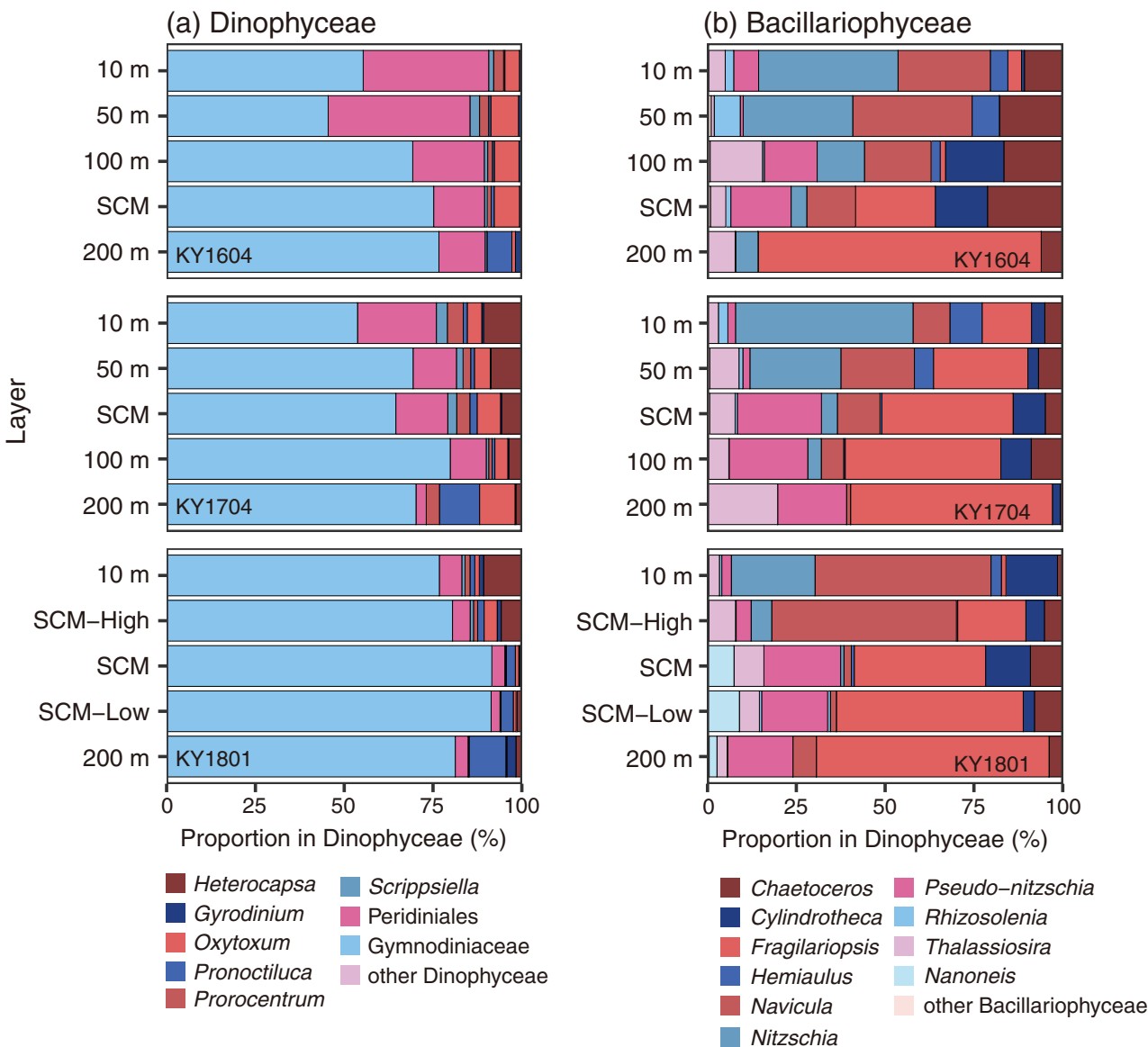

**Fig 5.** Vertical distribution of assemblages of (a) Dinophyceae and (b) Bacillariophyceae for KY1604 (top), KY1704 (middle), and KY1801 (bottom) cruises.

effects indicated that the target abundances are expected lower than the mean abundance, and the positive effects indicated the target abundances are expected higher than the mean abundance. When the additive effect of one parameter was observed wider range than the other parameter, the effects of that parameter was larger than the other parameters.

The effects of temperature showed a peak at 23–24˚C to the abundances of Bacillariophyceae, and at 20–21˚C to those of Dinophyceae (Fig 6). The response to salinity was consistent in all four groups, and the abundance decreased linearly with salinity (Fig 6). The response to latitude showed unimodal effects (Flagellates and Cryptophyceae) or monotonical negative effects with high latitude (Dinophyceae and Bacillariophyceae, Fig 6). The response to the nitrate concentration was similar among four groups, except Bacillariophyceae (Fig 6). The abundance of Bacillariophyceae decreased under a nitrate-depleted condition and showed a peak at sub-micromolar condition (Fig 6).

**Table 2. Corrected Akaike information criterion (AICc)-selected best GAM descriptions and deviance explained for every micro-size plankton abundance.**

| Class | Least-AICc models | Deviance explained (%) |
|---|---|---|
| Flagellates | | |
| (abundance) | $s(T) + s(S) + s(Lat) + s(logNO3) + \beta$ | 43.3 |
| Dinophyceae | | |
| (abundance) | $s(T) + s(S) + s(Lat) + s(logNO3) + s(Ro) + \beta$ | 47.1 |
| Bacillariophyceae | | |
| (abundance) | $s(T) + s(S) + s(Lat) + s(logNO3) + \beta$ | 33.4 |
| Cryptophyceae | | |
| (abundance) | $s(T) + s(S) + s(Lat) + s(logNO3) + s(Ro) + \beta$ | 33.9 |
| Prasinophyceae | | |
| (present–absent) | $s(S) + s(Lat) + s(logNO3) + s(Ro) + \beta$ | 27.4 |
| (abundance) | $s(S) + s(logNO3) + \beta$ | 34.3 |
| Dictyochophyceae | | |
| (present–absent) | $s(T) + s(S) + s(Ro) + \beta$ | 28.2 |
| (abundance) | $s(T) + s(Lat) + s(logNO3) + s(Ro) + \beta$ | 21.8 |
| Cyanophyceae | | |
| (present–absent) | $s(T) + s(Lat) + \beta$ | 34.0 |
| (abundance) | $s(S) + s(Lat) + \beta$ | 33.6 |

The $s$ denotes the smoothing function, and T, S, Lat, logNO3, Ro, and β denote water temperature, sea salinity, latitude, common logarithm transformed nitrate concentration, relative vorticity, and intercept, respectively.

The delta GAMs were applied to Dictyochophyceae, Prasinophyceae, and Cyanophyceae (Fig 7, Table 2). Different environmental parameters were selected (Table 2), and different environmental responses were sometimes shown between the present–absent GAMs and the abundance GAMs (Fig 7).

Temperature was not selected in the GAMs on Prasinophyceae presence and abundance (Fig 7A). Temperature was selected as explanatory variables in the cases of Dictyochophyceae and Cyanophyceae (only their presence, Fig 7). The presence of Dictyochophyceae showed a peak between 24 and 26˚C (Fig 7), and the temperature effect on their abundance was selected but was not significant ($p > 0.05$, Fig 7B). The presence probability of Cyanophyceae increased in warm waters (Fig 7C). The response to salinity was ambiguous to the presence probability when selected as explanatory variable (Fig 7A and 7B), but showed significantly negative effects to the abundances of Prasinophyceae and Cyanophyceae (Fig 7A and 7C). The latitude showed that the presence probabilities of Prasinophyceae and Cyanophyceae increased in the high-latitude area. On the other hand, the abundance of Dictyochophyceae was high in the low-latitude area, and the abundance of Cyanophyceae was ambiguous. Both the presence probability and abundance were high in nitrate-depleted water in the case of the Prasinophyceae (Fig 7A). The presence probability of Dictyochophyceae was also high in low-nitrate water, less than 1 μM. The nitrate concentration was not selected as the explanatory variable in the case of Cyanophyceae. The Ro was selected in the case of Dictyochophyceae, but the effects on their presence probability and abundance were not significant.

## Variations in assemblages based on PCoA

The PCoA, applied to class-level plankton assemblages, summarized approximately three quarters of variation in assemblages with first and second axes (Axis 1 and 2, respectively) based on the eigenvalues (Fig 8). Temperature, common logarithm nitrate concentration,

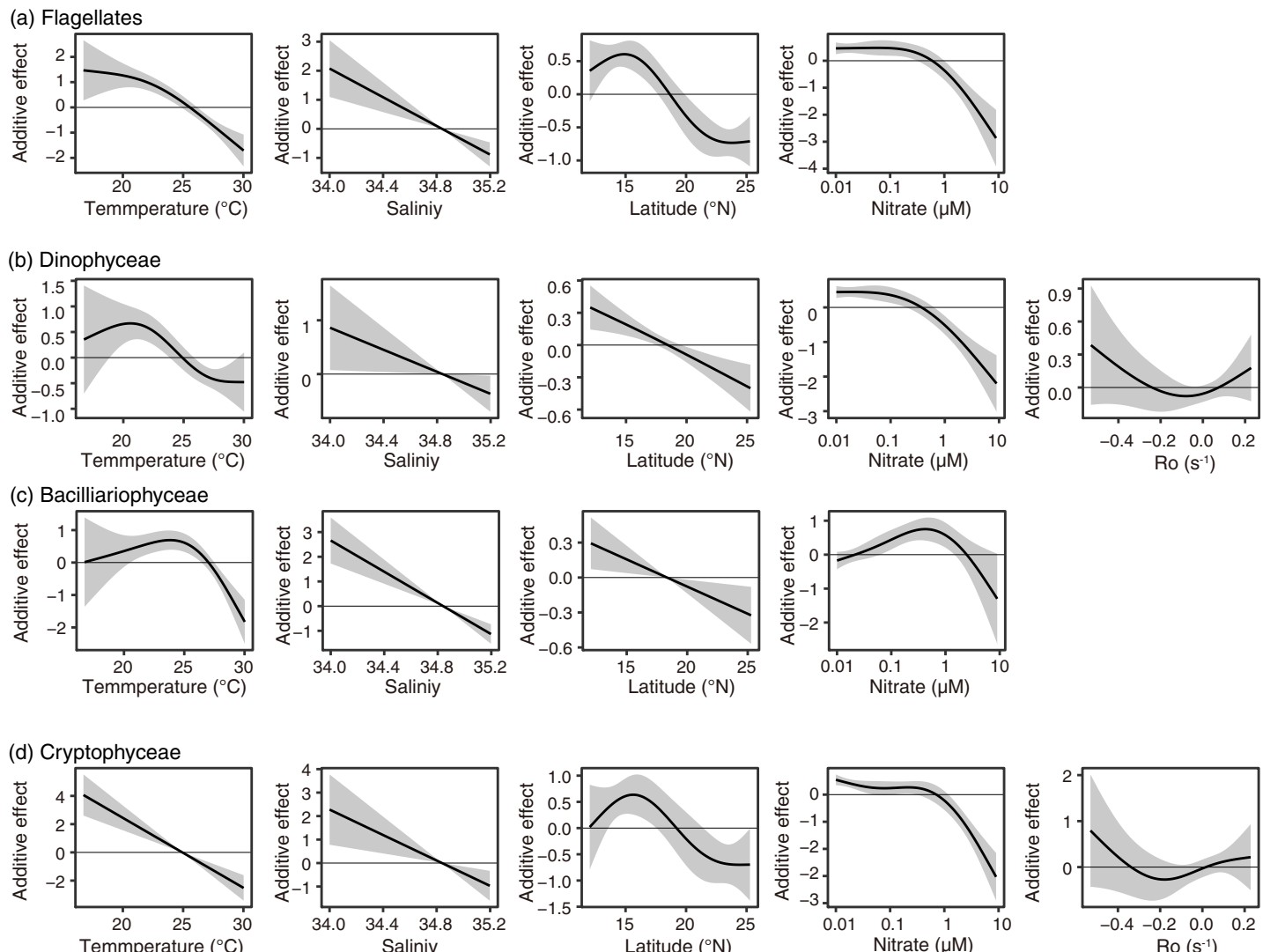

**Fig 6.** Partial effects of GAMs on (a) flagellates, (b) Dinophyceae, (c) Bacillariophyceae, (d) Cryptophyceae. The black lines are the smoothing terms, and shadows denote the 95% confidence intervals. The blank panels denote the parameters that were not selected in using the AIC selection.

varied significantly with the assemblages summarized in Axis 1 and 2 of class-level PCoA ($p < 0.05$); however, the other parameters (i.e., salinity, latitude, and Ro) were not significant. Axis 1 explained 57.6% of variations and Axis 2 explained 18.7% of variations, denoting that positive values were warm and low nitrate concentrations (Fig 8). The weighting average score of every class denoted that two rare classes, Haptophyceae (Hap) and Euglenophyceae (Eug), differed from the other classes (Fig 8). The other classes were plotted in similar positions and their score was positive along Axis 1. Along Axis 2, Cyanophyceae (Cya) and Dictyochophyceae (Dic) were scored positive, but the other classes, i.e., Bacillariophyceae (Bac), Dinophyceae (Din), Prasinophyceae (Pra), Cryptophyceae (Cry) and flagellates (Fla), were scored negative. The 95% ellipse of every sampling layer based on site scores of class-level PCoA denoted that plots of clusters of sites scored at the 200 m depth layer differed slightly from the that of other layers. In addition, the site scores at 10 m depth layer varied most widely (Fig 8).

The sample scores of the PCoA of the Dinophyceae showed the Dinophyceae community at 200 m depth differed from others, as well as the other PCoA (Fig 8). On the other hand, the

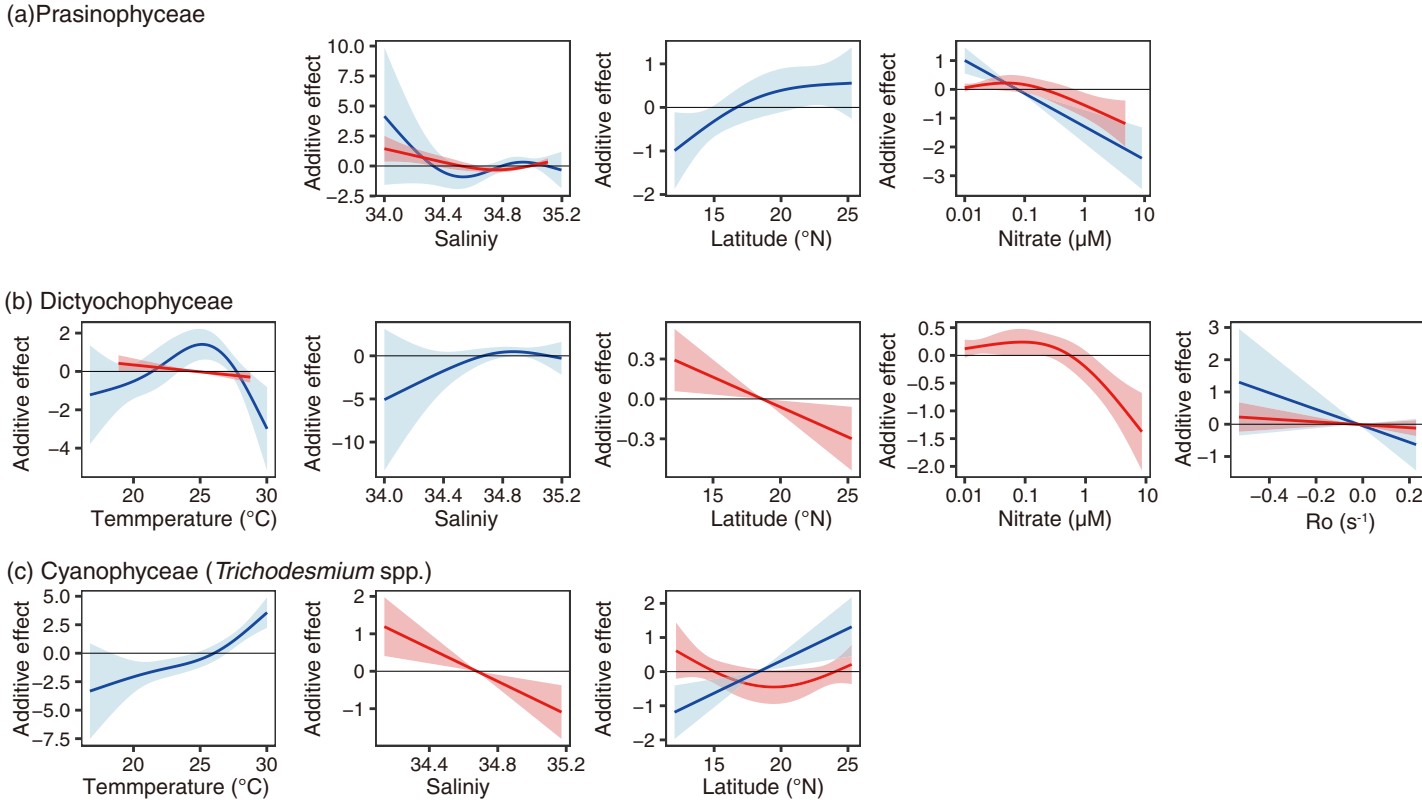

**Fig 7.** Partial effects of delta GAMs on (a) Prasinophyceae, (b) Dictyochophyceae, and (c) Cyanophyceae (*Trichodesmium* spp.). The blue line with blue shadow indicates the result of the present–absent GAM, and that of the red the result of the abundance GAM. The shadows denote the 95% confidential intervals.

scores of every genus, including Peridiniales (Peri) and Gymnodiniaceae (Gymn), plotted to quite similar positions along Axis 1. Along Axis 2, *Heterocapsa* (Hete), *Gyrodinium* (Gyro), *Pronoctiluca* (Pron), and Gymnodiniaceae plotted in the negative, and *Oxytoxum* (Oxyt), *Prorocentrum* (Proc), Scrippsiella (Scri), and Peridiniales plotted in the positive. Temperature and nitrate concentration were correlated significantly with the site scores.

The PCoA applied to abundance of the genus of Bacillariophyceae summarized 39.4% of variations with Axis 1 and 2. Temperature, salinity, and nitrate concentration significantly explained the variations. Axis 1 explained 24.8% of variation and Axis 2 explained 14.6% of variation, denoting that negative values plotted in the warm, less-saline, and nitrate-poor conditions (Fig 8). The sample scores denoted that the Bacillariophyceae community at 200 m depth differed from those at the other depths, as well as the class level and Dinophyceae community structure. The scores of every genus showed two clusters, one of which was consistent with *Rhizosolenia* (Rhi), *Navicula* (Nav), *Hemiaulus* (Hem), and *Nitzschia* (Nit), and the other was consistent with *Chaetoceros* (Cha), *Cylindrotheca* (Cyl), *Fragilariopsis* (Fra), *Nanoneis* (Nan), *Pseudo-nitzschia* (Pse), and *Thalassiosira* (Tha). The scores of samples at depths of 10 and 50 m plotted near the first groups.

## Discussion

Our study focused on micro-size plankton abundance and assemblages in the western NPSG. Our numerical abundance-based micro-size plankton community structure showed results similar to carbon-based results in the western NPSG [6] and numerical abundance-based

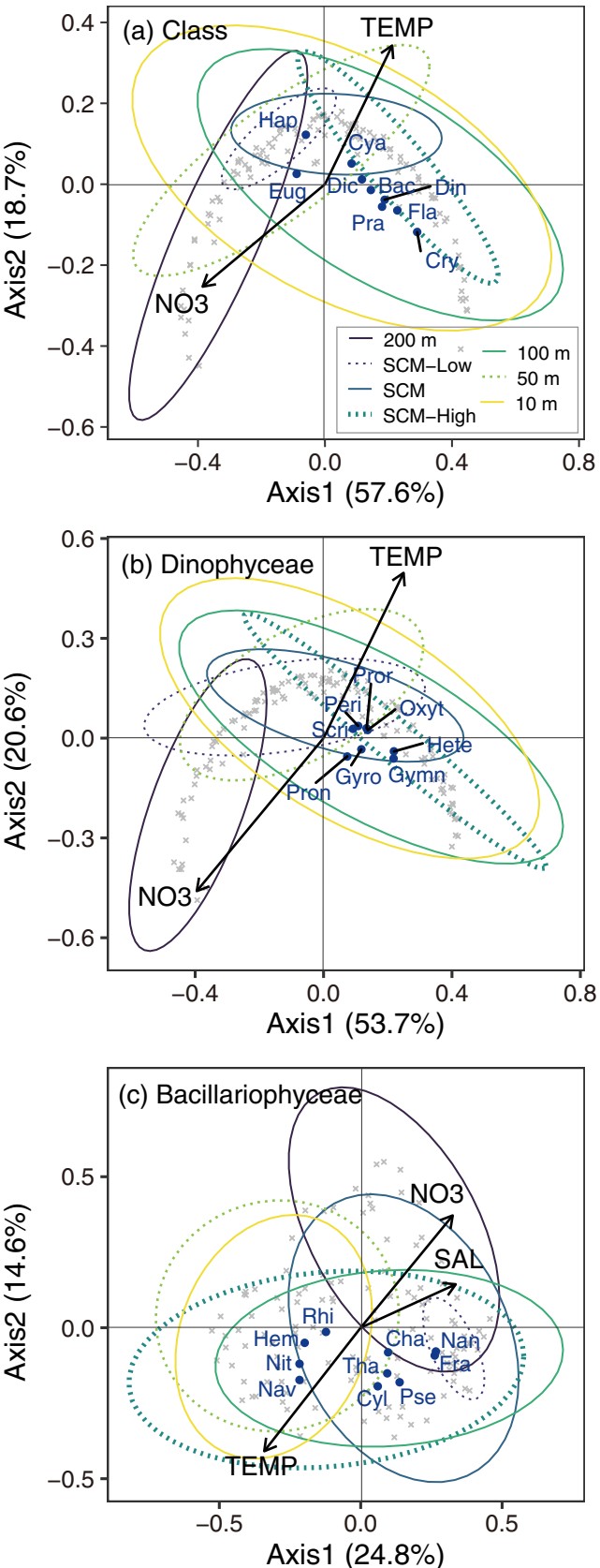

**Fig 8.** Triplots of the results on the PCoA of (a) class level, (b) those of the Dinophyceae, and (c) those of the Bacillariophyceae. The gray and blue plots are sample scores and target group scores, and the arrows are environmental parameters. The ellipses divided by color and line type denote plotted ranges of sample scores at every sampling layer regardless of the cruises. The abbreviations TEMP, SAL, and NO3 denote temperature, salinity, and nitrate concentrations, respectively. The abbreviations of target group (dark blue fonts) are shown in the main text.

results [16] conducted over three decades. Interestingly, comparing the results of this study and the observations of half a century ago [6, 16] indicated that the micro-size plankton community structure could be considered conservative in the western NPSG, whereas primary production in the western NPSG depicted slight increase [18]. Our investigation, however, could not determine the density of coccolithophores that make a significant contribution as the primary producer of this area [6, 16], with ~10% of carbon explained by coccolithophores [6]. The abundance of coccolithophores was between that of Bacillariophyceae and Dinophyceae [16]. Therefore, further investigation is necessary to improve understanding of the entire micro-size plankton community of this area.

The recent results of eukaryotic 18S rRNA gene sequence compositions based on metabarcoding technique also showed Dinophyceae are dominant >75% of eukaryotic community in the western NPSG [39]. The metabarcoding results [39] were consistent with our microscopic observations, with the exception of Dinophyceae which comprised a larger proportion of the community in metabarcoding results. This is likely due to Dinophyceae having larger nuclear genome sizes compared to other microplankton [40]. In addition, the contribution of other flagellates was remained very low in the metabarcoding technique results [39]. This observation suggests that while microscopic observation is a time consuming technique, it is still a necessary tool to understand the micro-size plankton community in the oligotrophic ocean; furthermore, this indicates that the metabarcoding technique should best be coupled with more traditional methods such as microscopic observations.

The contribution of micro-size plankton to primary production is low, as the contributions of >10 μm-Chl *a* concentration to the total Chl *a* concentration in the discrete samples were almost <10%. In addition, at every depth, flagellates and Dinophyceae were abundant. The dominant Dinophyceae in our study area (Peridiniales and Gymnodiniaceae) are considered heterotrophs [41–43]. In addition, most of the pico- and nano-size phytoplankton of this area are considered mixotrophs [44]. The proportion of autotrophic flagellates was not evaluated in this study. However, we did consider the micro-size plankton community as not comprising mainly primary producers, but were ranked as secondary or more higher producers. These predacious flagellates and Dinophyceae prey on pico-size plankton, which is abundant in this region. The Dinophyceae were detected from larval eel guts based on the metabarcoding technique [45]. Therefore, the microbial loop is suggested important for the biological production of this area, and micro-size plankton is the link between pico-size plankton and higher trophic animals.

The elevations of abundances in the less-saline water are the common feature, except Dictyochophyceae, shown in GAMs. The mechanism of high micro-size plankton abundance in the less-saline water was not explained in previous studies, but Shiozaki et al. [46] reported that active nitrogen fixation and primary production are observed in the less-saline (<34.2) water mass of the North Equatorial Current. In the less-saline water observed by Shiozaki et al. [46], *Trichodesmium* (and *R. intracellularis*) increased, similar to the findings of the present study. Therefore, we considered that the elevation of micro-size plankton in the less-saline water could be attributed to active nitrogen fixation. The importance of nitrogen fixation in this area is also pointed out in the results of the nitrogen isotope ratios of pelagic fish and squid muscle [47].

We could not detect the effects of meso-scale eddies on the micro-size plankton abundance (and also assemblages), whereas other studies [27–29] have reported that the phytoplankton abundance vary with the eddies. Several studies focused on the variations in chlorophyll *a* concentration, but micro-size plankton abundance and assemblages vary with meso-scale eddies, as reported by Davis and McGillicuddy [48]. Two reasons were considered for our results on the meso-scale eddies, namely 1) the difference in observation of oceanic basins, and 2) the limitations of our approach (observations and statistical models). First, previous studies had been conducted in the central and eastern NPSG [27–29], and the heterogeneity of plankton abundance and assemblages with the meso-scale eddies are observed rarely in the western NPSG. Second, our data sets were a mixture of observations of three cruises, of which the primary aims were investigation of the distribution of Japanese eel larvae. Well-organized observations would be necessary for evaluation of the effects of meso-scale eddies on micro-size plankton abundance and assemblages.

The heterogeneity of the plankton assemblages was another topic of our study. Among the dominant four groups at the class level, similar responses to salinity and latitude were observed in the GAMs. Habitat segregation was caused mainly by temperature and nitrate concentration among the four classes, based on the GAMs. Our results in this regard are consistent with the results of the PCoA. Comparing the response to temperature among four groups, Bacillariophyceae increased in abundance and proportion in the micro-zooplankton in warm water, and Cryptophyceae and flagellates increased in cold water. The response to the nitrate concentration indicated that Bacillariophyceae abundance increased at the nitracline, and the other three groups were increased in the nitrate-depleted waters. Collinearity was not high, but the relationship between temperature and nitrate concentration showed negative correlations; therefore, we rarely found warm and nitrate-rich condition nor cold and nitrate-poor condition at our study sites. We considered that these conflicted optimal conditions observed in these four classes could be attributed to the mixing of several genera, as shown in the PCoA of Bacillariophyceae.

Wei et al. [49] reported that Bacillariophyceae are more abundant than are the Dinophyceae in the western NPSG. Compared with micro-size plankton assemblages in the eastern NPSG [7], our results were similar at class level, but Bacillariophyceae abundance in the eastern NPSG was higher than that indicated in our study [7]. In the eastern NPSG, nitrogen-fixing diatom symbioses blooms are sometimes observed [10], and vertical migration of *Rhizosolenia* is observed [50]. During the KY1704 and KY1801 cruises, DDAs abundance was $<70$ host cells $L^{-1}$, and such a low abundance of DDAs could be the reason for low Bacillariophyceae abundance in our observations compared with the results from Wei et al. [49] and Venrick [7]. Therefore, we considered that Bacillariophyceae-dominant waters could be present in the western NPSG, but at a limited scale. Our results on the abundance of Bacillariophyceae are consistent with the findings of Hashihama et al. [51], who reported that the abundance of Bacillariophyceae was several tens of cells $L^{-1}$ at the surface of the western NPSG, except a diatom bloom in a cyclonic eddy at the northern edge of NPSG.

The results of PCoAs indicated that community structure at 200 m depth differed from those at other depths. At 200 m depth, nutrient concentration was at micromolar level, and the depth at which light intensity is 1% of the surface irradiance is reported as $<\sim150$ m in the western NPSG [46, 52]. The mean attenuation coefficient of photosensitive available radiation (PAR) was -0.0457 $m^{-1}$ during the other western NPSG cruise in 2019, which was conducted under the same project of this study, and it suggested that a compensation depth of 143 m for a light level of 0.1%, and a compensation depth of 95 m for a light level of 1%. Therefore, light is the potential limiting factor in the growth of micro-size phytoplankton at 200 m depth. Similar environmental conditions were observed at the SCM-Low layer (at 166 ± 8 m), but the

community structure differed from that at 200 m depth. Therefore, a unique micro-size plankton community was formed at 200 m depth compared with the other sunlit layers.

Latasa et al. [53] investigated the fine-scale vertical distribution of the phytoplankton community structure in the subtropical North Atlantic and concluded that nutrients and light are the most evident environmental variables for the difference in community structure. Some studies in the western NPSG indicated that the phytoplankton community in the deep ocean depended mainly on sinking from the sunlit surface [54, 55]. In this study, however, sinking from the shallower layers were not considered as the main factor of plankton community at 200 m depths because PCoA showed the micro-size plankton community at 200 m depth was unique. Some of the phytoplankton community at 200 m depth, such as *Fragilariopsis* could adapt to the low-light condition.

Comparing the heterogeneity of plankton assemblages among the sampling layers showed the plankton communities at 10 m depth were most heterogeneous based on PCoA, except those of Bacillariophyceae. At 10 m depth, the nitrate concentration was almost depleted, and the heterogeneous plankton assembly of this layer was not based on the heterogeneous nitrate supply from the deep water. We considered that nitrogen fixation could make the community heterogeneous in this area. Biotic nitrogen fixation occurs everywhere [56], but not homogenously in the western NPSG [13, 46, 52]. The micro-size plankton abundances were elevated in the less-saline and *Trichodesmium*-rich surface water. In addition, a part of unicellular nano-size Cyanobacteria has nitrogen fixation ability, and is distributed patchily in our observation area [57]. The difference in the diazotroph community links to the difference in primary production [58]. Therefore, we considered that the micro-size plankton community would be heterogeneous because both the quality and quantity of the nitrogen input via biological nitrogen fixation are heterogeneous at a depth of 10 m.

The micro-size plankton is not the major primary producers of the western NPSG, but they are one of the key groups in the ecosystems linked to the higher trophic organisms. Watanabe et al. [39] reported the direct importance of micro-size plankton for the larval eels growing up in the western NPSG. The other important role of the micro-size plankton of this area is the size-up of organic matter. The western equatorial Pacific is the nursery ground of the Skipjack tuna *K. pelamis*, and the primary production of that area are positively correlated to the stock (biomass) of *K. pelamis* in that area [59]. However, *K. pelamis* larvae are carnivores as well as the other tuna larvae [60]: they prey meso- and macro-size plankton. The pico- and nano size plankton is the main primary producers in the western NPSG, and thus the size-up processes of organic matter by micro-size plankton are essential for the higher trophic level organisms in the western NPSG. The surface micro-plankton abundance was considered to be controlled with nitrogen fixation in our study, the primary production in the less-saline and *Trichodesmium*-rich water is doubled comparing to the saline and *Trichodesmium*-poor water in the western NPSG [46], and the nitrogen isotope ratios of pelagic fish and squid muscle also indicates the importance of nitrogen fixation as the nitrogen source in the marine ecosystem of this area [47]; these facts indicated micro-size plankton links primary production to the higher trophic level organisms including commercially viable migratory fish in the ecosystems of this area.

## Conclusion

We investigated micro-size plankton communities and their habitats in the western NPSG based on microscopic observation. Heterotrophic plankton was indicated as abundant in the micro-size plankton community; therefore, microbial loops are considered important for biological production. The assemblages did not differ markedly in the euphotic layer. A unique

community was observed only at a depth of 200 m. Most heterogeneous communities of micro-size plankton were recorded at 10 m depth, suggesting that a heterogeneous nitrogen fixation amount and Cyanobacteria groups made the plankton community heterogeneous. In addition, abundant micro-size plankton was observed in surface waters, where diazotrophic micro-size Cyanophyceae (*Trichodesmium* spp.) are present in rich abundance. Accordingly, understanding the microbial loops and the abundance and community of the diazotrophs is a prerequisite to further evaluation of biological productivity in the western NPSG. We consider genetical approaches, including environmental DNA, as promising methods for understanding the productivity of the western NPSG, but the genetical approaches are partly compatible approaches of microscopic observations at this time.

## Acknowledgments

We thank the captains, crews, and scientists of RV *Kaiyo-Maru*, the Fisheries Agency of Japan, for their invaluable support with sampling. We also thank Kyoko Kawanobe for counting of the large phytoplankton.

## Author Contributions

**Conceptualization:** Taketoshi Kodama.

**Data curation:** Taketoshi Kodama.

**Formal analysis:** Taketoshi Kodama.

**Funding acquisition:** Daisuke Hasegawa.

**Investigation:** Taketoshi Kodama, Tsuyoshi Watanabe, Yukiko Taniuchi, Akira Kuwata, Daisuke Hasegawa.

**Methodology:** Taketoshi Kodama, Tsuyoshi Watanabe, Yukiko Taniuchi.

**Project administration:** Daisuke Hasegawa.

**Resources:** Tsuyoshi Watanabe, Yukiko Taniuchi.

**Software:** Taketoshi Kodama.

**Supervision:** Akira Kuwata, Daisuke Hasegawa.

**Validation:** Akira Kuwata.

**Visualization:** Taketoshi Kodama.

**Writing – original draft:** Taketoshi Kodama.

**Writing – review & editing:** Taketoshi Kodama, Tsuyoshi Watanabe, Yukiko Taniuchi, Akira Kuwata, Daisuke Hasegawa.

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
