## [Decision Letter · Decision Letter 0]

23 Feb 2021

PONE-D-20-37689

Micro-size plankton abundance and assemblies in the western North Pacific Subtropical Gyre under microscopic observation

PLOS ONE

Dear Dr. Kodama,

Thank you for submitting your manuscript to PLOS ONE. After careful consideration, we feel that it has merit but does not fully meet PLOS ONE’s publication criteria as it currently stands. Therefore, we invite you to submit a revised version of the manuscript that addresses the points raised during the review process.

The review that I have received indicates the usefulness of this manuscript to the area of research and has made several suggestions. A thorough revision is sought for further consideration.

We look forward to receiving your revised manuscript.

Kind regards,

Arga Chandrashekar Anil, Ph. D., D. Agr.,

Academic Editor

PLOS ONE

Journal Requirements:

Reviewers' comments:

Reviewer's Responses to Questions

**Comments to the Author**

1. Is the manuscript technically sound, and do the data support the conclusions?

Reviewer #1: Yes

2. Has the statistical analysis been performed appropriately and rigorously? 

Reviewer #1: Yes

3. Have the authors made all data underlying the findings in their manuscript fully available?

Reviewer #1: No

4. Is the manuscript presented in an intelligible fashion and written in standard English?

Reviewer #1: No

5. Review Comments to the Author

Reviewer #1: Summary: Authors called for a re-evaluation of the micro-size plankton community in the western NPSG to understand the hetero/homogeneity of their community structure. They collected plankton samples over 3 years (2016-2018) in the euphotic zone (<200 m) and analyzed via microscopy. It is commendable the amount of work that was conducted for this manuscript, using traditional methodology such as microscopy. The work is interesting in that the authors utilized several modeling approaches to elucidate drivers of specific microplankton populations in the NPSG. This work is comprehensive and thorough, despite its claiming that it was a “side project” during expeditions that had an alternative main focus. I believe that it will be a strong contribution to the field, with several key edits that I recommend below. Also, most of the manuscript is written in intelligible fashion and in standard English, except for a few choices of words (e.g. heterogenous) that are not correct, and also the use of parentheses to convey patterns of the opposite trend is confusing. In regards to these examples, the manuscript must be revised to be made more clear.

Major comments:

-This paper provides an overview of microplankton diversity and community structure in the western NPSG through microscopy. The method used is not necessarily novel, and I would have liked to see a combination or groundtruthing of more cutting edge techniques (e.g. DNA metabarcoding) with classic methodologies such as microscopy. I know that is not possible, so perhaps there could be a comparison with studies done with DNA in the area or similar areas to provide this context.

-Production measurements of microplankton would also provide greater understanding of the microplankton contribution and importance to ecosystem functioning. There are several references to this in the Discussion, but there must be historical measurements of carbon fixation in the area? This would also lean into the fisheries discussion that will set this study in a broader context.

-Many of the figures need more explanation in the Results. A few lines to explain the GAMs (Fig. 6-7), in terms of the basics (additive effect, the scales for each panel and how they differ), would be beneficial for the reader.

-I think that a Table showing the depths of the SCMs during each cruise (which is specified in L 226-228) would be more clear, or label the depths in the figures. It is also confusing that the SCM was 94 +/- 10 m in KY1704, but it is plotted beneath 100 m in the plots. I see you need to bin the samples this way because the samples weren’t always taken at the same depth, but it is a bit misleading to see a “depth profile” without the same range between the values on the y-axis. You could also try depth-integrating all the parameters to a specific light bin – 0-100m, 100 m to SCM, SCM to 200 m to break up the parameters around the Chlorophyll a profile. Where is your 1% surface light level? You could alternatively break it up according to light, but this may not be important for the heterotrophic folks – though they would likely be driven by their photosynthetic prey.

-Perhaps the SCM values in the Table could be combined with how many samples were taken at each depth and each cruise? You state that 127 total samples were taken – what is the breakdown (of the box plots in Figs. 3 and 4)? Also, are the environmental parameter sample count the same as the plankton sample count (in Fig. 2)?

Minor Comments:

Title: do they mean “assemblages” versus “assemblies”?

Abstract:

why is a re-evaluation necessary? Maybe one line expanding on what you describe in Introduction:

L31: Peridiniales?

L40: heterogeneous?

L40-43: major conclusions are lacking – the focus of the study is about reevaluating the microplankton, and the major conclusion from the abstract indicate nitrogen fixation is a large contributor.

-Introduction:

L47-49: Instead of using the parentheses, break up the pico vs micro comparison into another sentence.

L60: heterogeneous

Methods:

L96: were these measurements taken on an expedition that had a primary aim to investigate Japanese eel larvae? What does it mean that “other observations were not always organized well in terms of space and time”?

L115: fluorescence

L128: assemblages?

L168-L177: are there satellite altimetry showing sea level anomaly in this study region?

Results:

L234: nitracline is not the depth where there is 1 uM of nitrate concentration, itʻs the region of rapid change in nitrate concentration relative to depth. So for your cruises it looks like between 100-150 m.

Figure 1: the y-axis should be plotted on a linear scale, relative to the SCM. What are the SCM-High, SCM, and SCM-Low depths for each cruise? – applies to all the figures. Or have a table that tells the readers what the SCM depths are for each cruise?

L239-240: why do you think fluorometer values were overestimating the chl a concentration in the deeper layers?

Figure 4: It’s difficult to compare abundances, do they go from highest to less from left to right?

Figure 5: (A) Dinophyceae spelled with a y

L393 and 396 both start with “On the other hand…”

Figure 8b: Dinophyceae speeled with a y

L428: TEM should be TEMP

Discussion:

L466-468: Is this contribution of >10 um chl a concentration integrated from all depths sampled, or per depth? I would think that chl a concentrations and contributions would differ in varying light conditions (e.g. photoadaptation) so this must be evaluated accordingly with respect to light fields.

L515 and other places: When there is the polar opposite condition in parentheses, itʻs confusing. Please clarify.

L554-556, 562, 564, 574: heterogenous means “of foreign origin” and not the same as heterogeneous, which is the word I think you are trying to use.

L573: heterogenetic is not a word, I believe.

L579 to end: This ending sentence about eDNA for migratory fish comes out of nowhere – a few lines linking the study of microphytoplankton to migratory fish (like you do in Intro) in the Discussion would be beneficial.

Acknowledgments:

Any funding to acknowledge?

6. PLOS authors have the option to publish the peer review history of their article (what does this mean?). If published, this will include your full peer review and any attached files.

Reviewer #1: No

---

## [Author Response · Author response to Decision Letter 0]

16 Mar 2021

We copied the response letter which attached after the manuscript file. 

Response to Reviewer #1

Summary: Authors called for a re-evaluation of the micro-size plankton community in the western NPSG to understand the hetero/homogeneity of their community structure. They collected plankton samples over 3 years (2016-2018) in the euphotic zone (<200 m) and analyzed via microscopy. It is commendable the amount of work that was conducted for this manuscript, using traditional methodology such as microscopy. The work is interesting in that the authors utilized several modeling approaches to elucidate drivers of specific microplankton populations in the NPSG. This work is comprehensive and thorough, despite its claiming that it was a “side project” during expeditions that had an alternative main focus. I believe that it will be a strong contribution to the field, with several key edits that I recommend below. Also, most of the manuscript is written in intelligible fashion and in standard English, except for a few choices of words (e.g. heterogenous) that are not correct, and also the use of parentheses to convey patterns of the opposite trend is confusing. In regards to these examples, the manuscript must be revised to be made more clear. 

We appreciated the reviewer’s constructive comments. As indicated in the reviewer’s comment, this manuscript is the side project, and the data which we can use in this study were limited. For example, the observations were mostly conducted in night-time, and thus we cannot evaluate the light conditions. In this revision, we referred the results on the DNA metabarcoding approach collected in the same cruises but limited. We believe the further discussion using the metabarcoding results make our manuscript better in this revision.

Major comments:

-This paper provides an overview of microplankton diversity and community structure in the western NPSG through microscopy. The method used is not necessarily novel, and I would have liked to see a combination or groundtruthing of more cutting edge techniques (e.g. DNA metabarcoding) with classic methodologies such as microscopy. I know that is not possible, so perhaps there could be a comparison with studies done with DNA in the area or similar areas to provide this context. 

The DNA metabarcoding results in the limited cruise and stations were published in very recent (Watanabe et al. 2021 Scientific Reports); we can refer their results in the revised MS. The results of the DNA metabarcoding showed that Dinophytes are dominant group (over 90% in >3–10 µm) in the western NPSG. The Radiolaria and Copepods were following groups in >10 µm. The DNA metabarcoding approach also cannot clear the community structure of flagellates. We considered the results of metabarcoding are interesting, but we questioned its quantitative capability. Therefore, the morphological analyses were still effective for understanding the plankton community in the ocean. We added the comparison between metagenetic approaches and morphological approaches in the discussion (L. 478-488), and revised the conclusion (L625-628).

-Production measurements of microplankton would also provide greater understanding of the microplankton contribution and importance to ecosystem functioning. There are several references to this in the Discussion, but there must be historical measurements of carbon fixation in the area? This would also lean into the fisheries discussion that will set this study in a broader context.

We accepted this comment. We referred some studies on the primary production of this area (Shiozaki et al. 2013 and Yen & Lu 2016), and added the discussion the relationships between fisheries and micro-sized plankton (L593-611).

-Many of the figures need more explanation in the Results. A few lines to explain the GAMs (Fig. 6-7), in terms of the basics (additive effect, the scales for each panel and how they differ), would be beneficial for the reader.

We accepted this comment. We added (L366-373). 

-I think that a Table showing the depths of the SCMs during each cruise (which is specified in L 226-228) would be more clear, or label the depths in the figures. It is also confusing that the SCM was 94 +/- 10 m in KY1704, but it is plotted beneath 100 m in the plots. I see you need to bin the samples this way because the samples weren’t always taken at the same depth, but it is a bit misleading to see a “depth profile” without the same range between the values on the y-axis. You could also try depth-integrating all the parameters to a specific light bin – 0-100m, 100 m to SCM, SCM to 200 m to break up the parameters around the Chlorophyll a profile. Where is your 1% surface light level? You could alternatively break it up according to light, but this may not be important for the heterotrophic folks – though they would likely be driven by their photosynthetic prey. 

-Perhaps the SCM values in the Table could be combined with how many samples were taken at each depth and each cruise? You state that 127 total samples were taken – what is the breakdown (of the box plots in Figs. 3 and 4)? Also, are the environmental parameter sample count the same as the plankton sample count (in Fig. 2)? 

On these two comments, we added information on the depth of SCM in Table 1 (L101), and revised the figures as the order of the sampling depth. We are sorry, but we cannot measure light condition in these cruises because most of the observations were conducted in night-time. We added the information of light-conditions in the other cruises conducted in the same areas and in day-time (L560-564) for supporting that 200 m depth was not a sun-lit layer. 

Minor Comments:

Title: do they mean “assemblages” versus “assemblies”?

We revised as suggested. We revised in the text as well (L1).

Abstract:

why is a re-evaluation necessary? Maybe one line expanding on what you describe in Introduction:

We added (L17-20).

 L31: Peridiniales?

 L40: heterogeneous?

Sorry for careless mistakes. We revised (L31, and L40)

 L40-43: major conclusions are lacking – the focus of the study is about reevaluating the microplankton, and the major conclusion from the abstract indicate nitrogen fixation is a large contributor. 

We added as suggested (L23-24)

-Introduction:

 L47-49: Instead of using the parentheses, break up the pico vs micro comparison into another sentence. 

We added as suggested (L47-50).

 L60: heterogeneous

Sorry for typo; we revised (L61 and others).

Methods:

 L96: were these measurements taken on an expedition that had a primary aim to investigate Japanese eel larvae? What does it mean that “other observations were not always organized well in terms of space and time”?

We revised the sentence for clarifying the meaning (L97-99).

 L115: fluorescence

 L128: assemblages?

We revised (L124 and L137).

 L168-L177: are there satellite altimetry showing sea level anomaly in this study region?

Yes, but the sea level anomaly with satellites was 8 days composited data; therefore we chose the reanalysis data. Sea surface height anomaly was reflected in the geostrophic velocity anomalies in our study. We described the production ID (L186).

Results:

L234: nitracline is not the depth where there is 1 uM of nitrate concentration, itʻs the region of rapid change in nitrate concentration relative to depth. So for your cruises it looks like between 100-150 m. 

We revised the sentence (L244-245).

Figure 1: the y-axis should be plotted on a linear scale, relative to the SCM. What are the SCM-High, SCM, and SCM-Low depths for each cruise? – applies to all the figures. Or have a table that tells the readers what the SCM depths are for each cruise?

This is the comment on Figure 2. The SCM layer was different among the stations. When the average depth of SCM was put in the y-axis, the balance of the figure was bad. Therefore, we added the new table (Table 1), and describe the depth of SCM as well as depth of SCM-High and SCM-Low. The relationship between SCM and 100 m depth during the KY1704 cruise were revised. 

L239-240: why do you think fluorometer values were overestimating the chl a concentration in the deeper layers?

The reason was not cleared in our study, but the Falkowski and Keifer (1985) reviewed some reasons that in vivo fluorometer often does not adequately represent in situ chlorophyll a concentration. We added (L250).

Figure 4: It’s difficult to compare abundances, do they go from highest to less from left to right? 

We arranged the x-axis at the same scales (from 0 to 120) among the taxon but different orders (Figure 4).

Figure 5: (A) Dinophyceae spelled with a y

Sorry for typo. We revised (Figure 5). 

L393 and 396 both start with “On the other hand…”

We revised (L405).

Figure 8b: Dinophyceae speeled with a y

Sorry for typo. We revised. (Figure 8)

L428: TEM should be TEMP

We revised as suggested (L440).

Discussion:

 L466-468: Is this contribution of >10 um chl a concentration integrated from all depths sampled, or per depth? I would think that chl a concentrations and contributions would differ in varying light conditions (e.g. photoadaptation) so this must be evaluated accordingly with respect to light fields. 

This is the results of discrete samples and extracted chlorophyll a concentration. We consider the photoadaptation can be ignored in the analyses of extracted chlorophyll a concentration. We revised the sentence and clarify the sentence (L489-491).

 L515 and other places: When there is the polar opposite condition in parentheses, itʻs confusing. Please clarify. 

We revised (L539-540).

 L554-556, 562, 564, 574: heterogenous means “of foreign origin” and not the same as heterogeneous, which is the word I think you are trying to use.

 L573: heterogeneous is not a word, I believe. 

Sorry for typo. We revised (L580 and others).

 L579 to end: This ending sentence about eDNA for migratory fish comes out of nowhere – a few lines linking the study of microphytoplankton to migratory fish (like you do in Intro) in the Discussion would be beneficial. 

We added the discussion on the relationships between micro size plankton and migratory fish (L593-611).

Acknowledgments:

 Any funding to acknowledge? 

In the PLOS submission guideline, we found these sentences in the acknowledgements: Do not include funding sources in the Acknowledgments or anywhere else in the manuscript file. Funding information should only be entered in the financial disclosure section of the submission system.

---

## [Decision Letter · Decision Letter 1]

7 Apr 2021

PONE-D-20-37689R1

Micro-size plankton abundance and assemblages in the western North Pacific Subtropical Gyre under microscopic observation

PLOS ONE

Dear Dr. Kodama,

Thank you for submitting your manuscript to PLOS ONE. After careful consideration, we feel that it has merit but does not fully meet PLOS ONE’s publication criteria as it currently stands. Therefore, we invite you to submit a revised version of the manuscript that addresses the points raised during the review process.

The reviewer has recommended corrections to the text. Based on the opinion of the reviewer and my own assessment the manuscript can be considered for publication. Before the acceptance of the manuscript, I would like to see the corrected version.

We look forward to receiving your revised manuscript.

Kind regards,

Arga Chandrashekar Anil, Ph. D., D. Agr.,

Academic Editor

PLOS ONE

Journal Requirements:

Reviewers' comments:

Reviewer's Responses to Questions

**Comments to the Author**

1. If the authors have adequately addressed your comments raised in a previous round of review and you feel that this manuscript is now acceptable for publication, you may indicate that here to bypass the “Comments to the Author” section, enter your conflict of interest statement in the “Confidential to Editor” section, and submit your "Accept" recommendation.

Reviewer #1: All comments have been addressed

2. Is the manuscript technically sound, and do the data support the conclusions?

Reviewer #1: Yes

3. Has the statistical analysis been performed appropriately and rigorously? 

Reviewer #1: Yes

4. Have the authors made all data underlying the findings in their manuscript fully available?

Reviewer #1: Yes

5. Is the manuscript presented in an intelligible fashion and written in standard English?

Reviewer #1: Yes

6. Review Comments to the Author

Reviewer #1: Review #2

Micro-size plankton abundance and assemblages in the western North Pacific Subtropical Gyre under microscopic observation

In this version, the authors have addressed all of the comments of the reviewers to satisfaction. I have made few, mostly grammatical suggestions here:

L5 typo in the short title

L18-20 Edit first sentence to: “While primary productivity in the oligotrophic North Pacific Subtropical Gyre (NPSG) is changing, the micro-size plankton community has not been evaluated in the last 3 decades, prompting a re-evaluation.”

L23 Don’t need s in community structures.

L24 “The assemblages were consistent with those identified 4 decades previously.” Also 4 decades? Thought the first sentence indicated 3 decades.

L25 Dinophyceae were “the most numerically abundant, followed by Cryptophyceae and Bacillariophyceae (diatoms)”.

L42 Therefore, nitrogen fixation “may contribute” to the heterogeneity …

L49-50 In other words, energy fixed by primary production is more efficiently transferred to higher trophic organisms in micro-plankton dominated waters.

L61 is instead of as

L62 phytoplankton are “the dominant primary producers”;

L64 abundant but is it significant? What % of cell counts or Chl a?

L69 western NPSG is oligotrophic (no ‘the’)

L70 migratory fish?

L88-89 To understand the trophic structure and heterogeneity of this community.

L93 three cruises on the R/V …

L98-99 therefore, the observations for collections of seawater were limited to night-time collections.

Table 1 is very informative and much improved, thank you!

L121 – collected using … (no “by”)

L244 Nitrate concentration was over 1 uM of in water “where” the temperature was below …

L250 …layers as indicated “by” Falokowski …

L472 …primary production in the wester NPSG depicted slight increase.

L480 The metabarcoding results were consistent “with” our microscopic observation, with the exception of Dinophyceae which comprised a larger proportion of the community in metabarcoding results. This is likely due to Dinophyceae having largernuclear genome sizes compared to other microplankton.

L484-490 This observation suggests that while microscopic observation is a time consuming technique, it is still a necessary tool to understand the micro…..; furthermore, this indicates that the metabarcoding technique should best be coupled with more traditional methods such as microscopic observations.

7. PLOS authors have the option to publish the peer review history of their article (what does this mean?). If published, this will include your full peer review and any attached files.

Reviewer #1: No

---

## [Author Response · Author response to Decision Letter 1]

8 Apr 2021

Response to Reviewer

In this version, the authors have addressed all of the comments of the reviewers to satisfaction. I have made few, mostly grammatical suggestions here:

We thank the reviewer’s comments. I revised the manuscript as suggested. 

L5 typo in the short title

We revised from “Micro-sizes” to “Micro-size” (L5).

L18-20 Edit first sentence to: “While primary productivity in the oligotrophic North Pacific Subtropical Gyre (NPSG) is changing, the micro-size plankton community has not been evaluated in the last 3 decades, prompting a re-evaluation.”

We revised as suggested (L18-20).

L23 Don’t need s in community structures.

We revised from “community structures” to “community structure” (L22) as well as L556.

L24 “The assemblages were consistent with those identified 4 decades previously.” Also 4 decades? Thought the first sentence indicated 3 decades.

We revised from “3 decades” to “4 decades” of the first sentence (L19).

L25 Dinophyceae were “the most numerically abundant, followed by Cryptophyceae and Bacillariophyceae (diatoms)”.

We revised as suggested (L25).

L42 Therefore, nitrogen fixation “may contribute” to the heterogeneity …

We revised as suggested (L41).

L49-50 In other words, energy fixed by primary production is more efficiently transferred to higher trophic organisms in micro-plankton dominated waters.

We revised as suggested (L48-50).

L61 is instead of as

L62 phytoplankton are “the dominant primary producers”;

We revised “is the dominant primary producers” (L61-62).

L64 abundant but is it significant? What % of cell counts or Chl a?

We added “(> 108 cells m-2) in the surface mixed layer” with a new reference Scharek et al. 1999 DSRI (L64).

L69 western NPSG is oligotrophic (no ‘the’)

We revised as suggested (L69).

L70 migratory fish?

We revised as suggested (L70).

L88-89 To understand the trophic structure and heterogeneity of this community.

We revised as suggested (L88-89).

L93 three cruises on the R/V …

We revised as suggested (L93).

L98-99 therefore, the observations for collections of seawater were limited to night-time collections.

We revised as suggested (L98-99).

Table 1 is very informative and much improved, thank you!

We also thank the comments.

L121 – collected using … (no “by”)

We revised as suggested (L121).

L244 Nitrate concentration was over 1 uM of in water “where” the temperature was below 

We revised as suggested (L245).

L250 …layers as indicated “by” Falokowski …

We revised as suggested (L250).

L472 …primary production in the wester NPSG depicted slight increase.

We revised as suggested (L472).

L480 The metabarcoding results were consistent “with” our microscopic observation, with the exception of Dinophyceae which comprised a larger proportion of the community in metabarcoding results. This is likely due to Dinophyceae having largernuclear genome sizes compared to other microplankton.

We revised as suggested (L481-483).

L484-490 This observation suggests that while microscopic observation is a time consuming technique, it is still a necessary tool to understand the micro…..; furthermore, this indicates that the metabarcoding technique should best be coupled with more traditional methods such as microscopic observations.

We revised as suggested (L485-489).

---

## [Editor Report · Decision Letter 2]

12 Apr 2021

Micro-size plankton abundance and assemblages in the western North Pacific Subtropical Gyre under microscopic observation

PONE-D-20-37689R2

Dear Dr. Kodama,

We’re pleased to inform you that your manuscript has been judged scientifically suitable for publication and will be formally accepted for publication once it meets all outstanding technical requirements.

Kind regards,

Arga Chandrashekar Anil, Ph. D., D. Agr.,

Academic Editor

PLOS ONE
---

## [Editor Report · Acceptance letter]

15 Apr 2021

PONE-D-20-37689R2 

Micro-size plankton abundance and assemblages in the western North Pacific Subtropical Gyre under microscopic observation 

Dear Dr. Kodama:

I'm pleased to inform you that your manuscript has been deemed suitable for publication in PLOS ONE. Congratulations! Your manuscript is now with our production department. 

Kind regards, 

on behalf of

Professor Arga Chandrashekar Anil 

Academic Editor

PLOS ONE